# Bean and Nut Intake Were Protective Factors for Comorbid Hypertension and Hyperuricemia in Chinese Adults: Results from China Nutrition and Health Surveillance (2015–2017)

**DOI:** 10.3390/nu16020192

**Published:** 2024-01-06

**Authors:** Wei Piao, Shujuan Li, Qiya Guo, Xue Cheng, Xiaoli Xu, Liyun Zhao, Dongmei Yu

**Affiliations:** Key Laboratory of Trace Element Nutrition of National Health Commission, National Institute for Nutrition and Health Chinese Center for Disease Control and Prevention, Beijing 100050, China

**Keywords:** bean and nut, hypertension, hyperuricemia, comorbid, Chinese adults

## Abstract

This study aimed to describe the prevalence of comorbid hypertension and hyperuricemia (HH) and detected the dietary factors for HH in Chinese adults aged 18 to 64 years. All of the data were collected from the China Nutrition and Health Surveillance 2015–2017, with a stratified, multistage, random sampling method on a national scale. A total of 52,627 adult participants aged 18~64 years from the CNHS 2015–2017 were included in this study. HH was identified as SUA level cut-offs for males and females of 420 μmol/L and 360 μmol/L, respectively, with mean systolic blood pressure ≥140 mmHg and/or mean diastolic blood pressure ≥ 90 mmHg and/or received antihypertensive treatment within two weeks. The differences in HH prevalence between or among the subgroups were compared by the Rao–Scott chi-square test. The correlations between HH and covariates or metabolic factors were detected by a weighted two-level multivariate survey logistic regression. The total weighted sufficient intake ratios of beans and nuts, vegetables, and red meat were 59.1%, 46.6%, and 64.8%, respectively. The weighted prevalence of HH in the total participants was 4.7% (95% CI: 4.3–5.0%). The positive effects of bean and nut on HH were observed. The participants who had sufficient bean and nut intake showed lower risk for HH (for the total participants: OR = 0.734, 95% CI = 0.611–0.881). The prevalence of HH might have been a public health problem, and bean and nut intake might be a protective factor for HH in the Chinese population.

## 1. Introduction

Uric acid (UA) is a heterocyclic compound which is the final product of purine metabolism with 168 Dalton [1]. In the human body, UA is one of the antioxidant compounds that has a function in antioxidation, for example, protecting the DNA from degradation [2]. However, the negative effects of UA have been observed in previous studies, and the organ systems could be damaged by the elevated serum uric acid (SUA) level. The potential mechanism of negative effects is considered pro-oxidant and proinflammatory [3]. Different from other animals, low solubility of UA in the human metabolic system is due to a deficiency of activated uricase [4], and that could cause UA to develop accumulatively. According to the facts, SUA is mainly excreted through the kidney; if the rate of UA generation exceeds the rate of UA elimination, UA could be accumulated and result in high SUA levels. The relativities between high SUA levels and health problems, such as gout, stroke, cancer, obesity, urate kidney stones, and even cardiovascular diseases, have all been observed in previous studies [5,6,7,8,9]. Hyperuricemia is a status of high SUA concentration. Accumulating evidence has showed that people with hyperuricemia might not only be vulnerable to developing gout but might also suffer from non-infectious chronic diseases (NCDs), even in asymptomatic conditions [10]. In recent years, the prevalent trend of hyperuricemia has attracted more and more concerns in the field of public health. In mainland China, between 2000 and 2014, the prevalence of hyperuricemia was 13.3% [11], and then, it grew to 14.6% in 2015 [12]. Similar trends have also been observed in many other countries [13,14,15]. This suggests that hyperuricemia might have gradually become a global public health problem and should be paid more attention.

As independent risk factors for cardiovascular diseases (CVDs), the effects of hyperuricemia and hypertension have been observed in previous studies. Globally, 51% of stroke (cerebrovascular disease) and 45% of ischemic heart disease deaths are attributable to high systolic blood pressure [16]. The increasing prevalence of hypertension has been considered an important risk factor for CVDs, especially in developing countries [17]. Hypertension could increase the risk of negative impacts on various target organs, and this can lead to developing stroke, coronary heart disease, or heart failure [18]. In addition, hyperuricemia could also affect the progression of hypertension [19]. Some results in previous studies showed that the risks of subclinical atherosclerosis, CVDs, renal insufficiency, and all-cause mortality could be greatly increased by the synergistic interactions between hypertension and hyperuricemia [20,21]. The results from a Kailuan Study in Tangshan, China, showed that systolic blood pressure (SBP) and diastolic blood pressure (DBP) partially mediated the effect of SUA on incident CVDs, and the mediation effect was 57.64% for SBP and 46.27% for DBP [22]. Moreover, a cross-sectional survey conducted in Xinyang, China, in 2007 among a rural population aged 40–75 showed that the prevalence of hyperuricemia in hypertensive patients was 14.1%, and the rates of gender division in males and females were 21.5% and 10.2%, respectively [23].

Among the affect factors for hypertension and hyperuricemia, the dietary factor is a key one for the disease’s prevention, retardation, or even reversal of disease progression [24]. Dietary approaches to stop hypertension (DASH), a universally acknowledged diet pattern, was initially designed for preventing or controlling hypertension. Further studies on the effects of DASH showed that it might also have some potential mechanisms of action against risk factors for metabolic syndrome [25]. Another study on Mediterranean dietary approaches to stop hypertension for neurodegenerative delay (MIND) has been regarded as a novel healthy dietary pattern with huge benefits [26]. For hyperuricemia, diet had also been declared as a determinant that could exert either positive or negative effects on hyperuricemia development [27]. It has been confirmed that the content of purine in edibles plays a key role in fluctuations in UA levels in the circulatory system [28]. A previous study conducted with Chinese elderly populations showed that dietary factor was associated with the risk of hyperuricemia [29]. These consequences above suggest that hyperuricemia combined with hypertension should be regarded with serious concerns, and the effects of special diet patterns also need to be taken seriously in target populations, even in asymptomatic groups.

This study is based on the data from the China Nutrition and Health Surveillance (CNHS) 2015–2017 and aims to detect the correlative factors, including dietary factors, for comorbid hypertension and hyperuricemia (HH) in the Chinese adult population.

## 2. Materials and Methods

### 2.1. Data Source and Samples

The cross-sectional data of the study were collected from the CNHS 2015–2017 with representation at the national level. The details of sampling that have been described in a previous study [30]. And in this study, a total of 52,627 subjects aged 18~64 years old were screened by the criteria. The screening criteria are shown as follows: (1) the family or individual information of subjects was not integrated; (2) a food frequency questionnaire (FFQ) survey was not completely conducted, or the results were out of the logical ranges; (3) outliers that existed in the laboratory test or physical examination results. Written informed consent was obtained from all subjects and their guardians. 

### 2.2. Data Collection

Demographic characteristics and lifestyle habits of subjects were collected during a questionnaire interview conducted in a face-to-face manner, including the family and individual information. Height, weight, and blood pressure were measured by the height meter (TZG, Suzhou, China), the electronic weighing scale (G&G TC-200K, Changshu, China), and the Omron HBP-1300 electronic sphygmomanometer (Dalian, China), respectively. An FFQ was used to collect the data of dietary ingredient consumption.

The biochemical indicators, including HbA1c (glycosylated hemoglobin α-1-c), fasting blood glucose (FPG), total cholesterol (TC), low-density lipoprotein cholesterol (LDL-C), high-density lipoprotein cholesterol (HDL-C), triglycerides (TG), and SUA, were all tested by standard procedures [31].

### 2.3. Definition of Comorbid Hypertension and Hyperuricemia

Hyperuricemia was diagnosed according to the clinical diagnostic criteria, and the SUA level cut-off was 420 μmol/L for males and 360 μmol/L for females [32]. Hypertension was defined as mean systolic blood pressure ≥140 mmHg and/or mean diastolic blood pressure ≥ 90 mmHg and/or receiving anti-hypertension treatment within two weeks [33].

### 2.4. Dietary Intake Assessment

Dietary ingredients were divided into 6 kinds of food sources such as vegetable, fruit, milk, bean and nut, red meat, and alcohol. The intake of bean and nut, vegetable, fruit, and milk were assessed by the criteria of the Chinese Dietary Guidelines (2022): (1) sufficient vegetable intake was defined as a daily average intake ≥300 g; (2) sufficient fruit intake was defined as a daily average intake ≥200 g; (3) Sufficient milk intake was defined as a daily average intake ≥300 g. The quantity of milk was converted by protein content to 3.3 g/100 mL; (4) sufficient bean and nut intake was defined as a daily average intake ≥25 g; (5) the conditions of red meat intake were categorized as insufficient (daily average red meat intake < 18 g), moderate (18 g to 27 g), or excessive (>27 g) [34]; (6) alcohol consumption was categorized by international guidelines for monitoring alcohol consumption and related harm (WHO) as follows: never consumed (consume < 1 g of alcohol per day for both male and female), low-risk amount consumed (consume 1 to 40 g of alcohol per day for male or consume 1 to 20 g of alcohol per day for female), medium-risk amount consumed (consume 41 to 60 g of alcohol per day for male or consume 21 to 40 g of alcohol per day for female), or high and very high-risk amount consumed (consume ≥ 61 g of alcohol per day for male or consume ≥ 41 g of alcohol per day for female). The transfer coefficients of alcohol quantity were decided by the alcohol drink’s degree: the transfer coefficient of high-degree liquors was 50%; the transfer coefficient of low-degree liquors was 38%; the transfer coefficient of beers was 4%; the transfer coefficient of wines was 10% and the transfer coefficient of yellow rice or glutinous rice wine was 18%.

### 2.5. Covariates

Potential covariates were brought into the analysis: (1) gender; (2) urban and rural were categorized according to the region codes which were prescribed by the National Bureau of Statistics; (3) geographic regions were defined as east, central, and west according to the National Bureau of Statistics’ regulation, where east areas included Beijing, Tianjin, Shanghai, Hebei, Shandong, Fujian, Guangdong, Liaoning, Jiangsu, Zhejiang, and Hainan; central areas included Shanxi, Heilongjiang, Hubei, Jilin, Jiangxi, Hunan, Anhui, and Henan; west areas included Inner Mongolia, Gansu, Shaanxi, Guangxi, Guizhou, Ningxia, Chongqing, Sichuan, Yunnan, Tibet, Qinghai, and Xinjiang; (4) four age groups were established: 18–29, 30–39, 40–49, and 50–64; (5) educational situations were divided into three levels: low such as primary school or below, moderate such as junior school, and high such as high school or above; (6) four household income groups were established according to the Chinese Statistical Summary 2018: <65,141.7 CNY was identified as low, 65,141.7–193,494.9 CNY was identified as moderate, >193,494.9 CNY was identified as high, and others were unknown/unclear; (7) body mass index (BMI) was categorized as underweight, normal, overweight, and obese by the criteria of BMI < 18.5, 18.5 ≤ BMI < 24.0, 24.0 ≤ BMI < 28.0, and BMI ≥ 28.0; (8) smoking was classified as never smoked, formerly smoked, or currently smoke; (9) physical activity was categorized by the weekly total metabolic equivalent of task (MET): insufficient as MET < 600 and sufficient as MET ≥ 600 [35]; (10) the ADA 2010 criteria were used to diagnose diabetes mellitus, such as FPG level ≥ 7.0 mmol/L and/or HbA1c concentration ≥ 6.5% [36]; (11) according to the guidelines of the Chinese adult dyslipidemia prevention and treatment (2016 revised edition), dyslipidemia was identified by the criteria of total cholesterol ≥ 6.2 mmol/L, triglyceride ≥ 2.26 mmol/L, LDL ≥ 4.14 mmol/L, or HDL < 1.04 mmol/L [29]; (12) vegetarians were defined as those who consumed a plant-based diet with no eggs and dairy products.

### 2.6. Statistical Analysis

SAS version 9.4 software (SAS Institute Inc., Cary, NC, USA) was used for all data processing. To maintain the national representative, the weight of the sample was accessed by data from the China National Bureau Statistics. The PROC SURVEYFREQ procedure was used to calculate the demographic information of subjects. The weighted prevalence of HH was described as the case rate with a 95% confidence interval (95% CI), and the differences between or among the subgroups were compared by the Rao–Scott chi-square test. The influencing factors of HH were explored by the PROC SURVEYLOGISTIC procedure, and the results were presented as odds ratio (OR) and 95% CI by comparing them with the reference in each item. Sensitivity analysis was operated by 2 steps: The first step (model 1) was an unadjusted model. In this model, only dietary factors were included. The second step (model 2) was further adjusted for the indicators of demography, geography, and physiology. A *p*-value < 0.05 was considered to be statistically significant.

### 2.7. Quality Control

All of the staff members in this study were well trained and qualified by the China CDC project team, and all of the investigation work was conducted under strict supervision.

## 3. Results

### 3.1. Participant Characteristics

A total of 52,627 adult participants aged 18~64 years from the CNHS 2015–2017 were selected in this study, including 24,425 males and 28,202 females, and the weighted constituent ratios were 51.6% and 48.4% for males and females, respectively. The demographic and clinical characteristics of the study participants are summarized in Table 1. It shows that the weighted ratios for overweight and obesity in males (50.5%) were higher than in females (44.2%). And comparing with females, the incidences of diabetes mellitus and dyslipidemia were also observed at a higher level in males. In this study, the weighted ratios of HH were significantly higher in males (6.8%) than those in females (2.4%).

### 3.2. Dietary Intakes

For the dietary conditions, the total weighted sufficient intake ratios of beans and nuts, vegetables, and red meat were 59.1%, 46.6%, and 64.8%, respectively, and they were higher in males than in females. However, comparing with males, females in this study were more likely to have fruit and be vegetarian. The weighted sufficient intake ratio of fruits in females was 25.2%, even though it was at a relatively low level. And the weighted ratio of vegetarian in females was 6.2%. Significant differences in alcohol consumption between genders were observed. The weighted ratio of medium-risk and above alcohol consumption were 12.5% in males and 0.7% in females, respectively. The details are shown in Table 1.

### 3.3. Prevalence of HH among Participants

The weighted prevalence of HH in the total participants was 4.7% (95% CI: 4.3–5.0%). In different genders, the weighted prevalences of HH were 6.8% (95% CI: 6.2–7.5%) and 2.4% (2.1–2.6%). More details are shown in Table 2. In general, participants who were older, current smokers, heavier alcohol drinkers, consumed more red meat, and were in higher BMI groups showed a higher significant prevalence of HH. However, some discordant results were observed between the analyses in the total participants and categories of genders. For the vegetarian diet, a lower significant prevalence of HH was observed in the total participants and males but not in females. For red meat intake, significant differences were observed in total participant subgroups, but they were not observed in gender subgroups. Although no significant differences in the prevalence of bean and nut intake were detected in the total participants and male subgroup, the trends were similar to those in the female subgroup (the lower bean and nut intake subgroup had lower prevalence of HH). Meanwhile, the participants with diabetes mellitus or dyslipidemia showed a higher significant prevalence of HH in total participants and genders.

### 3.4. Associations between Dietary Factors and HH

Results for association between dietary factors and HH are presented in Table 3, Table 4 and Table 5 by survey logistic regression. Among the dietary correlative factors, alcohol consumption played a key role in HH. Irrespective of whether in the crude model (model 1) or in adjusted model (model 2), the negative effects of alcohol consumption on HH were observed in the total participants and genders, except for in the model 2 in females. The significant positive effects of a vegetarian diet were observed in the total participants (OR = 0.679, 95% CI = 0.462–0.996), but not in males or females. In our study, the positive effects of bean and nut intake were remarkable. It is obvious that the participants who had sufficient bean and nut intake showed lower risk for HH, even in model 2 (for the total participants, OR = 0.734, 95% CI = 0.611–0.881; for males, OR = 0.733, 95% CI = 0.587–0.917; for females, OR = 0.739, 95% CI = 0.567–0.963). Although the significant effects of red meat were not observed in all categories, similar trends also emerged.

## 4. Discussion

Hyperuricemia and hypertension are two NCDs that seriously threaten the cardiovascular health. Elevated SUA levels have been recognized to be associated with the risk of hypertension development [37,38]. It has been suggested that hyperuricemia combined with hypertension might cause some superimposed effects on human being health, and this needs to be paid close attention to. Meanwhile, the regulatory effects of diet on hyperuricemia and hypertension have also been confirmed. Appropriate dietary intake behaviors or adjustments in food ingredients could positively prevent or postpone the onset of NCDs, including hyperuricemia and hypertension. From the results of this study, the prevalences of hyperuricemia and hypertension are close to those in developed countries; meanwhile, the incidence of HH in the Chinese population and the positive effects of bean and nut on HH should also be paid more attention. 

Due to the growing morbidity rate, hyperuricemia has gradually become an important worldwide public health issue. Elevated serum uric acid levels have been confirmed as being associated with gouty arthritis and cognized as a risk for CVDs [28,39]. In addition, the association with increased risks of onset and progression of chronic and end-stage kidney disease in hyperuricemia patients was also observed [40]. Data from the National Health and Nutrition Examination Survey (NHANES) reported that the hyperuricemia prevalence in the United States (US) was 15.9% and 14.6% in 2007–2008 and in 2015–2016, respectively. A similar result was detected in the Japan national survey; the hyperuricemia prevalence in Japan was 13.4% in 2016–2017. In our study, the overall weighted hyperuricemia prevalence was 15.1% (21.2% in males; 8.5% in females), which was consistent with the findings of Liu et al. [11], and higher than Maloberti’s results [41]. This suggested that the risk level of hyperuricemia in the Chinese population was elevated to be close to that of developed countries. This also showed that hyperuricemia appeared to be more common in males, which was consistent with NHANES’ and Zhang’s results [42]. The lower hyperuricemia prevalence in females might be attributed to the effects of estrogen [43]. In a previous study, no significant difference in hyperuricemia prevalence between genders was present in the elderly age groups with decreased hormone levels [12]. Moreover, the lifestyle of females, especially the dietary pattern, was very different from the males [29]. It had been proven in several studies that red meat and alcohol consumption were two of the riskiest factors for hyperuricemia [44,45,46], and they were obviously lower in females than in males [12].

Hypertension is one of the most important risk factors for CVD, and there is a consensus that modifying hypertension is an effective measure for improving the quality of life. However, the prevalence of hypertension in both developed and developing countries is increasing dramatically, including in the Chinese population [14,47,48]. Data from US adults in the three NHANES cycles between 2013 and 2018 showed that about 33.0% of the US adult population had hypertension [48]. The 2010–2014 Korea National Health and Nutrition Examination Survey reported that hypertension prevalence in Korean adults in males and females was 34.6% and 30.8%, respectively [49]. Another nationally representative study covering about 1,320,555 adults was conducted in India. From this study, the crude prevalence of hypertension in India was 25.3% [50]. In China, a study was conducted by Liu et al. in Chongqing in southwest China that showed that the overall hypertension prevalence among adults was 23.9%, with 23.4% in males and 24.4% in females. In our study, the total weighted prevalence of hypertension was 23.5%, (26.8% in males and 19.9% in females). The weighted prevalence of hypertension in males was significantly higher than in females. Compared with the results of these studies, our results are basically consistent with Liu’s and the Indian study’s results, whereas they are lower than the US and Korea in contemporaneity. Although the prevalence of hypertension in the Chinese population was lower than that in developed countries, it was still at a relatively high level that needs to be taken seriously, as the prevalence rate was nearly 5%, increasing compared with the data from the 2002 China National Nutrition and Health Survey (NNHS) [51].

In recent years, more attention has been paid to the negative effects of HH. A previous study found that the level of SUA was also associated with target organ damage in patients with hypertension [17]. Data from a Chinese report revealed that hyperuricemia and hypertension had accounted for more than 40% of all deaths and become the leading cause of death in China [52]. In terms of the mechanism, elevated SUA levels could induce nephropathy and vascular changes and thus lead to the development of hypertension; on the contrary, hypertension could lead to glomerular hypoxia for microvascular dysfunction and then inhibit the excretion of SUA, inducing hyperuricemia [22]. It was suggested that the temporal relationship between them might be bidirectional, and both should be given equal attention. Dietary amelioration is the cross point of the methods for preventing hyperuricemia and hypertension. In previous studies, risk factors for hyperuricemia and hypertension such as genetics, gender, age, residence location, physical inactivity, obesity, hyperlipidemia, prediabetes, and diabetes had been verified [53,54]. A reduced salt intake, moderation of alcohol intake, vegetarian diets, and fish oil supplementation were documented as having positive effects on the direction of blood pressure [24], and red meat intake, alcohol consumption, vegetarian diets, and bean and nut intake were included in the influencing factors for hyperuricemia [53]. In our study, associations between food composition and HH were detected. In the total participants, bean and nut intake, alcohol consumption, and vegetarian diets were significantly associated with HH in model 1, and the associations were not changed when further adjusted in model 2. For gender, significant associations between HH and bean and nut intake and alcohol consumption were observed in males; however, only bean and nut intake were detected as significant associations with HH in females. The rational interpretation might be the physiological characteristics and the dietary factors. For the physiological characteristics, fluctuations in estrogen concentration could be one of the causes of the differences between genders. The potential mechanisms might be the negative effects of E2 on the metabolism of SUA, and with increasing age, the prevalence of hyperuricemia between genders converged [55]. 

Dietary patterns, as the changeable factors, play a pivot role in the prevention of hyperuricemia and hypertension. It has been confirmed that DASH or MIND dietary patterns have positive effects on lowering blood pressure; however, some ingredients are also beneficial to hyperuricemia. In the present study, higher bean and nut intakes were found to have positive effects on lowering the risk for HH. Beans and nuts are cognized as nutritious foods. Both are rich in monounsaturated fatty acids (MUFAs) and polyunsaturated fatty acids (PUFAs), dietary fiber, vitamins, minerals, polyphenols, and antioxidants. PUFAs, especially ω-3 PUFAs, could improve endothelial function [56,57]. For SUA, unsaturated fatty acids, especially long-chain unsaturated fatty acids, could inhibit urate transporter 1 in the renal tubules, and then regulate the urate in the crude urine reabsorbed into the blood [58]. Moreover, some studies inferred that mineral intake was negatively correlated with hyperuricemia and hypertension [59,60]. Taking dietary magnesium (Mg), for instance, is negatively correlated with C-reactive protein levels, while the latter is positively correlated with hyperuricemia [61]. For hypertensive subjects, some studies have demonstrated that appropriate mineral supplementation, such as potassium, magnesium, or calcium, is beneficial to hypertension amelioration [62,63]. And these nutrients mentioned above are rich in the compositions of beans and nuts.

To the best of our knowledge, this study was the first to describe the prevalence of HH and detect the dietary correlative factors for HH in the Chinese population which was nationally representative. Without a doubt, alcohol consumption was still the greatest risk factor for HH due to the conclusive mechanisms. And a novel protective factor, bean and nut intake, was detected to be the common dietary protective factor for the two disorders. Simultaneously, our study had some limitations that needed to be addressed. Firstly, due to the cross-sectional design, this study cannot well elucidate the causal and temporal correlation between bean and nut intake and HH. Future prospective cohort studies and more biological experiments are needed to confirm the findings and unravel the exact mechanisms. Secondly, the diet data were collected by an FFQ to evaluate the dietary intake during the whole year, which might cause recall bias. Thirdly, due to the overall design of the CNHS (2015–2017), several covariables, such as genes, morbid state, hormone levels, emotional status, or other remaining unmeasurable factors, were not considered in this study.

## 5. Conclusions

The prevalence of HH should be taken seriously as it might be a public health problem, especially in the male population. The present study found that in the Chinese population, bean and nut intake might be a protective factor for HH, and the negative effects of alcohol intake on HH were verified.

## Figures and Tables

**Table 1 nutrients-16-00192-t001:** Demographic, dietary, and clinical characteristics of all participants in this study.

	Overall		Male		Female	%	*p*-Value *
	N	% ^¶^	N	% ^¶^	N		
**Total**	**52,627**	**100**	**24,425**	**51.6**	**28,202**	**48.4**	
**Bean and nut intake**							
Insufficient	23,919	40.9	10,614	39.1	13,305	42.9	<0.0001
Sufficient	28,708	59.1	13,811	60.9	14,897	57.1	
**Vegetable intake**							
Insufficient	28,559	53.4	12,914	51.7	15,645	55.2	<0.0001
Sufficient	24,068	46.6	11,511	48.3	12,557	44.8	
**Fruit intake**							
Insufficient	42,642	78.4	20,441	81.7	22,201	74.8	<0.0001
Sufficient	9985	21.6	3984	18.3	6001	25.2	
**Milk intake**							
Insufficient	51,657	98.0	24,053	39.1	27,604	42.9	0.0878
Sufficient	970	2.0	372	60.9	598	57.1	
**Red meat intake**							
Insufficient	15,026	26.5	5503	20.0	9523	33.5	<0.0001
Moderate	4463	8.7	1805	7.3	2658	10.2	
Excessive	33,138	64.8	17,117	72.7	16,021	56.4	
**Alcohol consumption**							
Never	37,810	69.4	11,883	47.9	25,927	92.3	<0.0001
Low risk	10,712	23.9	8746	39.7	1966	7.0	
Medium risk	1513	2.6	1346	4.6	167	0.4	
High and very high risk	2592	4.2	2450	7.9	142	0.3	
**Vegetarian**							
No	49,760	94.9	23,366	95.9	26,394	93.8	<0.0001
Yes	2867	5.1	1059	4.1	1808	6.2	
**Residence location**							
Urban	21,143	52.3	9576	53.5	11,567	51.2	0.0110
Rural	31,484	47.7	14,849	46.5	16,635	48.8	
**Geographic region**							
East	20,612	44.5	9495	45.1	11,117	43.8	0.3167
Central	14,868	30.3	6889	29.8	7979	30.9	
West	17,147	25.2	8041	25.1	9106	25.3	
**Age group (years)**							
18~29	5642	26.6	2493	26.4	3149	26.7	0.6759
30~39	7796	23.3	3534	23.6	4262	22.9	
40~49	14,399	25.7	6595	25.4	7804	26.1	
50~64	24,790	24.5	11,803	24.6	12,987	24.3	
**Education level**							
Low	22,316	29.0	8247	22.5	14,069	36.1	<0.0001
Moderate	17,984	35.3	9658	38.4	8326	32.0	
High	12,327	35.7	6520	39.1	5807	32.0	
**Household income**							
Low	34,917	60.4	16,324	60.4	18,593	60.5	0.0544
Moderate	8490	20.8	3914	21.3	4576	20.2	
High	929	2.5	430	2.7	499	2.3	
Unknown	8291	16.3	3757	15.6	4534	17.1	
**BMI**							
Wasting	2067	5.1	856	4.7	1211	5.6	<0.0001
Normal	24,294	47.4	11,133	44.8	13,161	50.2	
Overweight	18,548	33.1	8907	35.4	9641	30.7	
Obese	7718	14.3	3529	15.1	4189	13.5	
**Smoking**							
Never	35,717	67.3	8327	38.4	27,390	97.9	<0.0001
Former	14,142	28.4	13,471	53.4	671	1.7	
Current	2768	4.4	2627	8.1	141	0.4	
**Physical activity**							
Insufficient	29,962	64.0	12,984	60.9	16,978	67.2	<0.0001
Sufficient	22,665	36.0	11,441	39.1	11,224	32.8	
**Diabetes mellitus**							
No	48,487	94.0	22,381	93.6	26,106	94.5	0.0076
Yes	4140	6.0	2044	6.4	2096	5.5	
**Dyslipidemia**							
No	31,724	61.7	13,197	53.9	18,527	70.1	<0.0001
Yes	20,903	38.3	11,228	46.1	9675	29.9	
**Hyperuricemia**							
No	45,633	84.9	19,958	78.8	25,675	91.5	<0.0001
Yes	6994	15.1	4467	21.2	2527	8.5	
**Hypertension**							
No	35,743	76.5	15,917	73.2	19,826	80.1	<0.0001
Yes	16,884	23.5	8508	26.8	8376	19.9	
**Hypertension and Hyperuricemia**							
No	49,739	95.3	22,546	93.2	27,193	97.6	<0.0001
Yes	2888	4.7	1879	6.8	1009	2.4	

* *p*-value indicates statistical significance of the differences between genders. Constituent ratios are weighted. ^¶^ The values of polytomous variables may not sum to 100% due to rounding. Abbreviation: BMI—body mass index.

**Table 2 nutrients-16-00192-t002:** Weighted prevalence of HH among participants.

	Total			Males			Females		
	%	95% CI	*p*-Value *	%	95% CI	*p*-Value *	%	95% CI	*p*-Value *
**Total**	4.7	(4.3–5.0)		6.8	(6.2–7.5)		2.4	(2.1–2.6)	
**Residence location**									
Urban	4.9	(4.4–5.5)	0.1151	7.2	(6.2–8.2)	0.1771	2.4	(2.0–2.8)	0.7726
Rural	4.4	(3.9–4.8)		6.4	(5.6–7.1)		2.3	(2.0–2.7)	
**Area of the country**									
East	5.1	(4.5–5.8)	0.0627	7.4	(6.4–8.4)	0.2255	2.7	(2.2–3.1)	0.1362
Central	4.4	(3.7–5.0)		6.6	(5.4–7.7)		2.1	(1.7–2.6)	
West	4.2	(3.6–4.8)		6.1	(5.1–7.1)		2.2	(1.7–2.6)	
**Age (years)**									
18~29	2.5	(1.7–3.3)	<0.0001	4.1	(2.8–5.5)	<0.0001	0.8	(0.1–1.6)	<0.0001
30~39	3.6	(2.9–4.2)		5.9	(4.6–7.1)		1.0	(0.6–1.5)	
40~49	5.3	(4.6–6.1)		8.2	(7.0–9.4)		2.4	(1.9–2.9)	
50~64	7.3	(6.7–7.9)		9.2	(8.3–10.1)		5.3	(4.6–5.9)	
**Education level**									
Low	5.0	(4.4–5.7)	0.2220	7.5	(6.2–8.7)	0.4752	3.4	(2.9–3.9)	<0.0001
Moderate	4.8	(4.3–5.4)		6.9	(6.0–7.8)		2.2	(1.7–2.7)	
High	4.2	(3.4–5.0)		6.4	(5.1–7.6)		1.4	(0.8–2.0)	
**Household income**									
Low	4.8	(4.3–5.2)	0.4067	6.9	(6.2–7.7)	0.2277	2.5	(2.2–2.8)	0.4267
Moderate	4.8	(3.9–5.7)		7.5	(5.9–9.1)		1.9	(1.4–2.3)	
High	3.3	(1.8–4.9)		4.4	(1.8–7.0)		2.0	(0.7–3.3)	
Unknown	4.1	(3.1–5.2)		5.8	(4.5–7.2)		2.5	(1.4–3.6)	
**BMI**									
Wasting	1.5	(0.1–3.0)	<0.0001	0.9	(0.1–1.6)	<0.0001	2.1	(0.0–4.7)	<0.0001
Normal	1.7	(1.4–1.9)		2.6	(2.1–3.0)		0.8	(0.6–1.0)	
Overweight	6.2	(5.4–6.9)		8.8	(7.6–10.0)		2.9	(2.4–3.4)	
Obese	12.2	(10.7–13.8)		16.6	(14.1–19.1)		7.1	(5.8–8.4)	
**Smoking**									
Never	3.7	(3.3–4.1)	<0.0001	6.9	(5.8–7.9)	0.0009	2.4	(2.1–2.6)	0.6889
Former	6.1	(5.3–6.9)		6.2	(5.4–7.0)		2.8	(1.1–4.6)	
Current	10.4	(7.9–12.9)		10.7	(8.2–13.3)		3.7	(0.0–8.6)	
**Physically active**									
Insufficient	4.7	(4.3–5.2)	0.5658	7.3	(6.4–8.2)	0.0323	2.2	(1.9–2.6)	0.2563
Sufficient	4.5	(4.0–5.1)		6.0	(5.2–6.9)		2.6	(2.1–3.1)	
**Diabetes mellitus**									
No	4.3	(4.0–4.7)	<0.0001	6.5	(5.9–7.1)	<0.0001	2.1	(1.8–2.3)	<0.0001
Yes	9.7	(8.1–11.4)		11.7	(9.2–14.2)		7.3	(5.6–9.0)	
**Dyslipidemia**									
No	2.3	(2.0–2.6)	<0.0001	3.6	(3.0–4.2)	<0.0001	1.3	(1.0–1.5)	<0.0001
Yes	8.5	(7.7–9.2)		10.6	(9.5–11.7)		4.9	(4.2–5.7)	
**Bean and nut intake**									
Insufficient	5.0	(4.4–5.6)	0.1143	7.3	(6.2–8.4)	0.2355	2.8	(2.3–3.2)	0.0106
Sufficient	4.4	(4.0–4.9)		6.5	(5.7–7.3)		2.1	(1.8–2.4)	
**Vegetable intake**									
Insufficient	4.3	(3.8–4.8)	0.0231	6.4	(5.5–7.3)	0.2035	2.1	(1.7–2.5)	0.0965
Sufficient	5.1	(4.6–5.6)		7.2	(6.4–8.1)		2.6	(2.2–3.0)	
**Fruit intake**									
Insufficient	4.7	(4.3–5.1)	0.7301	6.8	(6.1–7.4)	0.6788	2.3	(2.0–2.6)	0.6510
Sufficient	4.5	(3.6–5.4)		7.1	(5.5–8.8)		2.5	(1.7–3.3)	
**Milk intake**									
Insufficient	4.7	(4.3–5.1)	0.2027	6.9	(6.2–7.5)	0.207	2.4	(2.1–2.6)	0.8373
Sufficient	3.4	(1.6–5.1)		4.3	(1.2–7.5)		2.6	(0.6–4.6)	
**Red meat intake**									
Insufficient	3.9	(3.4–4.4)	0.0007	5.9	(4.8–7.1)	0.1372	2.6	(2.1–3.1)	0.3871
Moderate	3.6	(2.7–4.6)		5.9	(4.0–7.8)		1.9	(1.1–2.7)	
Excessive	5.1	(4.6–5.6)		7.2	(6.4–7.9)		2.3	(1.9–2.7)	
**Alcohol consumption**									
Never	3.5	(3.2–3.8)	<0.0001	5.5	(4.7–6.2)	<0.0001	2.4	(2.1–2.7)	0.0907
Low risk	6.1	(5.1–7.0)		6.8	(5.7–7.9)		1.7	(1.0–2.3)	
Medium risk	9.7	(6.8–12.7)		10.4	(7.2–13.5)		1.9	(0.2–3.7)	
High and very high risk	12.7	(10.6–14.8)		13.0	(10.8–15.2)		4.0	(0.8–7.2)	
**Vegetarian**									
No	4.8	(4.4–5.2)	0.0050	6.9	(6.3–7.6)	0.0388	2.4	(2.1–2.7)	0.2862
Yes	2.9	1.9–3.9		4.4	2.5–6.3		1.8	0.9–2.7	

* *p*-value indicates statistical significance of the differences among subgroups for each factor. Abbreviation: HH—comorbid hypertension and hyperuricemia; BMI—body mass index; 95% CI—95% confidence interval.

**Table 3 nutrients-16-00192-t003:** Association between dietary factors and HH in the total participants.

	Model 1				Model 2			
	OR	(95% CI)	*F*	*p*-Value	OR	(95% CI)	*F*	*p*-Value
**Bean and nut intake**								
Insufficient	Ref.		8.97	0.0029	Ref.		11.00	0.0010
Sufficient	0.767	(0.645–0.913)			0.734	(0.611–0.881)		
**Vegetable intake**								
Insufficient	Ref.		3.54	0.0604	Ref.		2.52	0.1133
Sufficient	1.176	(0.993–1.392)			1.147	(0.968–1.361)		
**Fruit intake**								
Insufficient	Ref.		0.02	0.8828	Ref.		0.04	0.8463
Sufficient	1.019	(0.792–1.311)			1.025	(0.796–1.322)		
**Milk intake**								
Insufficient	Ref.		0.88	0.3474	Ref.		0.44	0.5089
Sufficient	0.77	(0.447–1.328)			0.818	(0.450–1.487)		
**Red meat intake**								
Insufficient	Ref.		2.09	0.1249	Ref.		1.86	0.1565
Moderate	0.865	(0.627–1.195)			0.889	(0.639–1.237)		
Excessive	1.119	(0.934–1.34)			1.135	(0.943–1.368)		
**Alcohol consumption**								
Never	Ref.		72.07	<0.0001	Ref.		19.37	<0.0001
Low risk	1.802	(1.502–2.161)			1.293	(1.045–1.600)		
Medium risk	2.972	(2.083–4.242)			1.678	(1.118–2.520)		
High and very high risk	4.014	(3.285–4.905)			2.544	(1.997–3.240)		
**Vegetarian**								
No	Ref.		3.94	0.0478	Ref.		4.09	0.0436
Yes	0.679	(0.462–0.996)			0.659	(0.439–0.988)		

Model 1 was an unadjusted model; model 2 was further adjusted for gender, residence location, area of the country, age, education level, household income, body mass index (BMI), smoking, physical active, diabetes mellitus, and dyslipidemia. Abbreviation: OR—odds ratio; 95% CI—95% confidence interval; F—value of F test.

**Table 4 nutrients-16-00192-t004:** Association between dietary factors and HH in males.

	Model 1				Model 2			
	OR	(95% CI)	*F*	*p*-Value	OR	(95% CI)	*F*	*p*-Value
**Bean and nut intake**								
Insufficient	Ref.		4.40	0.0365	Ref.		7.46	0.0065
Sufficient	0.795	(0.642–0.986)			0.733	(0.587–0.917)		
**Vegetable intake**								
Insufficient	Ref.		1.23	0.2676	Ref.		1.58	0.2090
Sufficient	1.119	(0.917–1.365)			1.138	(0.930–1.393)		
**Fruit intake**								
Insufficient	Ref.		0.32	0.5723	Ref.		0.12	0.7302
Sufficient	1.087	(0.813–1.453)			0.949	(0.706–1.276)		
**Milk intake**								
Insufficient	Ref.		1.08	0.2989	Ref.		1.24	0.2665
Sufficient	0.668	(0.311–1.433)			0.637	(0.287–1.413)		
**Red meat intake**								
Insufficient	Ref.		0.79	0.4536	Ref.		1.48	0.2293
Moderate	0.935	(0.637–1.372)			0.925	(0.624–1.372)		
Excessive	1.115	(0.888–1.400)			1.182	(0.930–1.501)		
**Alcohol consumption**								
Never	Ref.		26.99	<0.0001	Ref.		20.20	<0.0001
Low risk	1.289	(1.029–1.614)			1.350	(1.071–1.701)		
Medium risk	2.028	(1.413–2.909)			1.793	(1.177–2.732)		
High and very high risk	2.635	(2.110–3.291)			2.692	(2.095–3.459)		
**Vegetarian**								
No	Ref.		2.60	0.1078	Ref.		3.01	0.0835
Yes	0.677	(0.421–1.089)			0.640	(0.386–1.061)		

Model 1 was an unadjusted model; model 2 was further adjusted for gender, residence location, area of the country, age, education level, household income, body mass index (BMI), smoking, physical active, diabetes mellitus, and dyslipidemia. Abbreviation: OR—odds ratio; 95% CI—95% confidence interval; F—value of F test.

**Table 5 nutrients-16-00192-t005:** Association between dietary factors and HH in females.

	Model 1				Model 2			
	OR	(95% CI)	*F*	*p*-Value	OR	(95% CI)	*F*	*p*-Value
**Bean and nut intake**								
Insufficient	Ref.		7.46	0.0065	Ref.		5.04	0.0252
Sufficient	0.733	(0.587–0.917)			0.739	(0.567–0.963)		
**Vegetable intake**								
Insufficient	Ref.		1.58	0.2090	Ref.		1.30	0.2547
Sufficient	1.138	(0.930–1.393)			1.171	(0.892–1.537)		
**Fruit intake**								
Insufficient	Ref.		0.12	0.7302	Ref.		1.42	0.2345
Sufficient	0.949	(0.706–1.276)			1.295	(0.845–1.986)		
**Milk intake**								
Insufficient	Ref.			0.2665	Ref.		0.35	0.5567
Sufficient	0.637	(0.287–1.413)	1.24		1.272	(0.569–2.843)		
**Red meat intake**								
Insufficient	Ref.		1.48	0.2293	Ref.		0.38	0.6833
Moderate	0.925	(0.624–1.372)			0.830	(0.504–1.366)		
Excessive	1.182	(0.930–1.501)			1.034	(0.752–1.423)		
**Alcohol consumption**								
Never	Ref.		20.20	<0.0001	Ref.		0.68	0.5661
Low risk	1.350	(1.071–1.701)			0.781	(0.487–1.254)		
Medium risk	1.793	(1.177–2.732)			0.624	(0.22–1.772)		
High and very high risk	2.692	(2.095–3.459)			1.144	(0.351–3.724)		
**Vegetarian**								
No	Ref.		3.01	0.0835	Ref.		1.54	0.2145
Yes	0.717	(0.406–1.266)			0.693	(0.388–1.237)		

Model 1 was an unadjusted model; model 2 was further adjusted for gender, residence location, area of the country, age, education level, household income, body mass index (BMI), smoking, physical active, diabetes mellitus, and dyslipidemia. Abbreviation: OR—odds ratio; 95% CI—95% confidence interval; F—value of F test.

## Data Availability

The data presented in this study are non-public.

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
