# Peer review of "Bean and Nut Intake Were Protective Factors for Comorbid Hypertension and Hyperuricemia in Chinese Adults: Results from China Nutrition and Health Surveillance (2015–2017)"

_nutrients, 2024, doi:10.3390/nu16020192_

Round 1

Reviewer 1 Report

Comments and Suggestions for Authors

In recent years, the rising trend of hyperuricemia and hypertension has raised increasing concerns, and it should indeed receive more attention, as stated by the authors, with which I completely agree. The increasing prevalence of hypertension is considered an important risk factor for cardiovascular diseases (CVDs), and previous study results indicate that the risk of overall mortality can significantly increase due to the synergistic interactions of hypertension and hyperuricemia. Among the influencing factors of hypertension and hyperuricemia, dietary factors play a crucial role in disease prevention, slowing down, or even reversal. In this meticulously conducted study, a total of 52,627 Chinese adults aged 18–64 were selected based on criteria. The consumption of dietary components, including vegetables, fruits, milk, alcohol, red meat, beans, and nuts, was collected using the FFQ method. It is a very precise and well-executed work; I only noticed minor errors, such as the lack of explanation for the abbreviation BMI under Table 1, and the need for correction of the %¶ above. Explanation of abbreviations (HH, BMI, CI) is missing under Tables 2, and explanation of abbreviations (OR, CI, F) is missing under Tables 3, 4, and 5. References are missing in several places, for example: "It had been proven in several studies that red meat and alcohol consumptions were the two of the riskiest factors for hyperuricemia [45]" (more references could be inserted here if multiple studies confirmed these associations). In previous studies, risk factors for hyperuricemia and hypertension, such as gene, gender, age, residence location, physical inactivity, obesity, hyperlipidemia, prediabetes, diabetes had been verified (references are missing here). PUFAs, especially ω-3 PUFAs, could improve endothelial function (reference needed). Moreover, some studies inferred that minerals intake was negatively correlated with hyperuricemia and hypertension (references are needed). Thank you!

Author Response

Responses:

Dear reviewer, thank you for your comments and suggestions. All the explanations for the abbreviations that you pointed out have been added following the corresponding tables. For the references, we have supplemented the corresponding references. Thank You!

Reviewer 2 Report

Comments and Suggestions for Authors

The manuscript is well written, and the study described in a proper way. The large population sample and the detailed statistical analysis render the manuscript of potential interest. However, some minor aspects may be addressed by the authors: 

1. Given the descriptive nature of the study, the conclusion should be attenuated or balanced by acknowledging the limits of the study. 

2. It is not clear why prevalence of hypertension has been included in the aims of the study, given the fact that the analysis is primarily focused on prevalence and correlates with hyperuricemia.

Author Response

  1. Given the descriptive nature of the study, the conclusion should be attenuated or balanced by acknowledging the limits of the study.

Response 1:

Dear reviewer, thank you for your comments and suggestions. For the limited of the study, the conclusion should be attenuated or balanced, we revised the Conclusions section.” The present study found that, in Chinese population, bean and nut intake might be a protective factor for HH, and the negative effects of alcohol intake on HH was verified.” Thank You!

  1. It is not clear why prevalence of hypertension has been included in the aims of the study, given the fact that the analysis is primarily focused on prevalence and correlates with hyperuricemia.

Response 2:

Dear reviewer, thank you for your comments and suggestions. Besides hypertension, the prevalence of hyperuricemia is growing rapidly in recent years, including Chinese population, and it nearly to be a public health problem. Results from several studies showed that hypertension and hyperuricemia might have synergistic actions on CVDs. Meanwhile, the effects of dietary factors on hypertension and hyperuricemia had been verified, respectively. Based on these findings, we supposed that some of the dietary factors could affect hypertension and hyperuricemia simultaneously. Therefore, in this study, we aimed to detect the effects of dietary factors on the comorbid hypertension and hyperuricemia. For the objective of this study, the prevalences of hypertension, hyperuricemia and the comorbid hypertension and hyperuricemia were all that we care about. Thank You!
